# Prediction of Histological Grades and Ki-67 Expression of Hepatocellular Carcinoma Based on Sonazoid Contrast Enhanced Ultrasound Radiomics Signatures

**DOI:** 10.3390/diagnostics12092175

**Published:** 2022-09-08

**Authors:** Yi Dong, Dan Zuo, Yi-Jie Qiu, Jia-Ying Cao, Han-Zhang Wang, Wen-Ping Wang

**Affiliations:** 1Department of Ultrasound, Zhongshan Hospital, Fudan University, Shanghai 200032, China; 2Precision Health Institute, GE Healthcare China, Shanghai 201203, China

**Keywords:** hepatocellular carcinoma (HCC), histological grades, contrast enhance ultrasound (CEUS), radiomics, Ki-67 expression

## Abstract

Objectives: Histopathological tumor grade and Ki-67 expression level are key aspects concerning the prognosis of patients with hepatocellular carcinoma (HCC) lesions. The aim of this study was to investigate whether the radiomics model derived from Sonazoid contrast-enhanced (S-CEUS) images could predict histological grades and Ki-67 expression of HCC lesions. Methods: This prospective study included 101 (training cohort: *n* = 71; validation cohort: *n* = 30) patients with surgical resection and histopathologically confirmed HCC lesions. Radiomics features were extracted from the B mode and Kupffer phase of S-CEUS images. Maximum relevance minimum redundancy (MRMR) and least absolute shrinkage and selection operator (LASSO) were used for feature selection, and a stepwise multivariate logit regression model was trained for prediction. Model accuracy, sensitivity, and specificity in both training and testing datasets were used to evaluate performance. Results: The prediction model derived from Kupffer phase images (CE-model) displayed a significantly better performance in the prediction of stage III HCC patients, with an area under the receiver operating characteristic curve (AUROC) of 0.908 in the training dataset and 0.792 in the testing set. The CE-model demonstrated generalizability in identifying HCC patients with elevated Ki-67 expression (>10%) with a training AUROC of 0.873 and testing AUROC of 0.768, with noticeably higher specificity of 92.3% and 80.0% in training and testing datasets, respectively. Conclusions: The radiomics model constructed from the Kupffer phase of S-CEUS images has the potential for predicting Ki-67 expression and histological stages in patients with HCC.

## 1. Introduction

Hepatocellular carcinoma (HCC) is the most common malignant tumor of the liver, with a 5-year survival rate of less than 20% [1]. Currently, surgical resection is the first curable treatment choice for most patients diagnosed with HCC. However, the prognosis is still poor after surgical resection, mainly due to recurrence and metastasis [2,3].

Previous studies showed that histological grade, which represents the biological behavior of tumors, is one of the significant prognostic factors that correlates with the recurrence of HCC after surgery [4]. Compared with low-grade HCC, high-grade HCC is associated with a worse survival rate. The 5-year tumor-specific survival for Edmondson-Steiner grades I–II and III–IV were 81% and 18%, respectively, after curative resection in HCC patients without portal vein tumor thrombosis. Thus, the histological grade may provide valuable prognostic information for HCC lesions.

Moreover, concerning the prognosis of patients with HCC lesions, Ki-67 protein expression level is another independent indicator of tumor growth rate and poor prognosis in HCC [5]. Those with aggressive HCC with high levels of Ki-67 expression showed very poor overall survival (OS) and disease-free survival (DFS) after surgery. The Ki-67 expression reflects the degree of tumor proliferation activity and is significantly correlated with tumor grades. A high expression level of Ki-67 is associated with a higher tumor grade and mortality [6].

Clinically, the accurate prediction of Ki-67 protein expression level and histological grade of the tumor may help in the selection of appropriate and personalized treatment. For example, preoperative transhepatic arterial chemotherapy and embolization (TACE) treatment is always recommended in highly aggressive HCC patients [2], which finally improves survival and outcome in patients. Currently, HCC histological grades and Ki-67 protein expression levels can only be identified by histopathological evaluation and immunohistochemistry testing after surgical resection. Conventional imaging techniques, such as ultrasound, computed tomography (CT), or magnetic resonance (MR) imaging, cannot preoperatively determine the histopathologic grades or degree of Ki-67 expression [7,8]. Therefore, preoperative and noninvasive assessment of histopathological tumor grades and Ki-67 protein expression level in HCC lesions is vital to guide personalized treatment strategies in clinical practice.

As a noninvasive, convenient, and real-time imaging method, ultrasound is recommended as the first-line imaging tool for the detection and diagnosis of liver tumors. Sonazoid is a hepatic-specific contrast agent with a long Kupffer phase. Contrast-enhanced ultrasound (CEUS) with Sonazoid may be helpful for better characterizing focal liver lesions [9]. Studies have also evaluated the correlation between CEUS features and histological grades of HCC. According to previous reports, CEUS allows for the evaluation of microvascular perfusion in tumors, which has been correlated with the histological grade of HCC lesions [10,11,12,13]. However, little is known about the potential of Sonazoid CEUS in predicting histopathological grads and Ki-67 protein expression levels in HCC lesions.

Current radiomics approaches based on ultrasound images involve numerous advanced, quantitative, high-throughput features. Previous studies have reported that tumor characteristics at the genetic and cellular levels can be reflected by predictive, diagnostic, and prognostic models developed with radiomics signatures [14,15,16]. Several studies have aimed to investigate the potential value of radiomics approaches in preoperative prediction of the histopathological grades and Ki-67 protein expression level in HCC [17,18,19]. However, most of them mainly focused on the feasibility of computed tomography (CT) and magnetic resonance (MR) images [20].

The purpose of our study was to construct preoperative and noninvasive radiomic models based on the Kupffer-phase features of Sonazoid enhanced CEUS images and to predict histopathological grade and Ki-67 protein expression level of HCC lesions.

## 2. Materials and Methods

### 2.1. Study Design

In this prospective study, we constructed radiomics models based on Kupffer-phase features from Sonazoid CEUS (S-CEUS) images (Figure 1). Two sets of radiomic features were extracted from B-mode ultrasound images (BM-Model) and contrast-enhanced ultrasound images (CE-Model). Patients were assigned to training and testing groups with a ratio of 7:3. Patients with an earlier admission date were assigned to the training group, and those with a later date, to the testing group. Prediction models were constructed on a training dataset, and the predictive performance of BM-Model and CE-Model was evaluated and compared in the testing cohort. A threshold value of 10% was set for stratifying Ki-67 protein expression levels [6]. We aimed to identify early versus late stages of HCC and lower (<10%) versus elevated (>10%) levels of Ki-67 in HCC lesions.

### 2.2. Patients

This prospective study was approved by the institutional review board of our institution (ID: B2021-041, approval date: 18 January 2021). Informed consent was waived before the ultrasound examination. The procedure followed was in accordance with the Declaration of Helsinki.

The inclusion criteria were: (1) patients diagnosed with focal liver lesions and suspected to be HCC; (2) patients who underwent preoperative B mode ultrasound (BMUS) and S-CEUS examinations one week before surgery; (3) patients with no previous treatment, such as radiofrequency ablation, microwave ablation, chemotherapy treatment, etc.; and (4) patients’ final diagnosis were confirmed by surgical resection and histopathological results.

The exclusion criteria were: (1) patients with suspected hepatic lesions that could not be clearly visualized on BMUS scan; (2) patients who were under 18 years, pregnant, or had a contraindication to Sonazoid contrast agent; (3) patients with incomplete clinical information or lack of final immunohistochemical diagnosis; and (5) patients with multiple focal liver lesions.

Patients were sorted by admission dates and assigned to training and testing datasets with a ratio of 7:3; the training dataset consisted of patients admitted earlier than those in the testing dataset.

### 2.3. Contrast-Enhanced Ultrasound Examination

S-CEUS examinations were performed by 2 radiologists with at least 10 years of liver CEUS experience, with a Siemens Acuson Sequoia machine (Mountain View, CA, USA) equipped with a 5 C-1 convex array transducer. All patients fasted for at least 8 h before S-CEUS examinations. The BMUS scan was first performed to locate the suspected HCC lesion. Then, a dose of 0.5 mL of Sonazoid was injected as a contrast agent, followed by a 5 mL saline solution flush. CEUS enhancement features were recorded and analyzed according to current WFUMB guidelines [9]. S-CEUS data were stored in DICOM format. Kupffer phase was defined as the period starting 5 min after the injection of Sonazoid. A single frame best representing the tumor and its peripheral liver parenchyma 5 min after the injection of contrast agent was sampled for radiomic analysis.

### 2.4. Histopathological Examination

The histopathological examination was performed by an experienced pathologist with more than 15 years of experience, who was blinded to the clinical and radiological results. The resected specimens were fixed with 10% paraformaldehyde solution, embedded in paraffin, and cut into 4 μm thick sections for either hematoxylin–eosin (HE) staining or immunohistochemical identification of Ki-67 protein expression. Ki-67 protein expression was considered positive when the cell nuclei were stained brown to yellow. Immunoreactive cells were classified as low Ki-67 expression (≤10% immunoreactivity) or high Ki-67 expression (>10% immunoreactivity) [6]. The major histological grade of HCC was recorded according to the Edmondson -Steiner classification. Tumors with Edmondson-Steiner grades I and II were classified into a low-grads group, while Edmondson-Steiner grade III and IV HCCs were classified into a high-grade group [4].

### 2.5. Radiomic Analysis

#### 2.5.1. Image Preprocessing and Tumor Segmentation

Kupffer-phase CEUS and BMUS images of HCC lesions with pixel values transformed from RGB pseudo-colored to grayscale were collected for quantitative radiomic analysis. Normalization and denoising algorithms were applied to adjust for interpatient variations in CEUS examinations and to remove image noise. The ultrasound imaging system used in this study produced images that simultaneously displayed CEUS and BMUS modes. HCC lesions were manually segmented on either CEUS or BMUS depending on the visibility of the lesion using 3DSlicer (version 4.8.1, slicer.org) by ultrasound radiologist 1 with 5 years of experience; then, the segmentation was copied to the other half using a symmetry-based algorithm. This ensured that the tumor’s shape-related features in CEUS and BMUS were identical for a single patient. Initial tumor segmentations were confirmed by ultrasound radiologist 2 with more than 15 years of experience. In the case of an unclear tumor–liver boundary, the tumor was recorded with an ‘unclear boundary’ and the consensus reached by both radiologists was used as the final segmentation of that tumor. Both radiologists were blinded to the pathological results at the time of segmentation and confirmation.

To evaluate the radiomic features’ segmentation-dependence, images from 20 randomly selected patients were resegmented by radiologist 1, and feature extraction was repeated. Radiomic signatures in the repetition were tested against features in the original extraction using the intraclass correlation coefficient (ICC). Test/retest analysis using concordance correlation coefficient (CCC) was calculated to evaluate feature stability. Features that suffered poor reproducibility or stability (defined as ICC < 0.75, CCC < 0.90) were omitted from further analysis.

#### 2.5.2. Preprocessing and Radiomic Features Extraction

Quantitative radiomic features were extracted using the open-source PyRadiomics package according to IBSI standards. Histogram-based gray level co-occurrence matrix (GLCM), gray level dependency matrix (GLDM), gray level size zone matrix (GLSZM), neighboring gray tone difference matrix (NGTDM), and gray level run length matrix (GLRLM) features were extracted from the Laplacian of Gaussian, wavelet transform, and local binary pattern filter-derived images. A bin width of 2 was used for histogram-based feature extraction. Two sets of radiomic features were extracted from each patient: (1) radiomic features of the grayscale image and (2) radiomic features of the Kupffer CEUS image. Z-score normalization was applied to the training dataset, then to the testing set, using the mean and standard deviation of the training data.

#### 2.5.3. Radiomic Feature Selection and Model Development

The model was developed on the training dataset and then tested for performance on the testing dataset. For simplicity of prediction, HCC stages were further classified into an advanced group consisting of stage III patients and a mild group consisting of the rest.

To coarsely narrow down the scope of valuable features, a subset of features having the most relevance with the prediction target and least correlation among themselves was identified by applying the minimal redundancy maximum relevance (MRMR) algorithm, which is a filter-based feature selection method that ranks feature relevance and intrafeature correlation using F statistics and Pearson’s correlation coefficient. The MRMR-selected subset of features was fitted to the prediction target using multivariate logistic regression. To optimize the prediction performance of the logistic regression, L1 regularization was applied to attenuate the feature coefficients, with its penalizing factor tuned with five-fold internal cross-validation. By setting the coefficient of feature least contributing to the final prediction to zero, L1-regularized logistic regression achieved further feature selection. Ten-fold cross-validation was applied in the training dataset to finalize candidate features.

The finally selected features were applied in a stepwise multivariate logistic regression minimizing the Akaike information criterion (AIC) to formulate the final prediction model.

Based on the modality of the images from which the radiomic features were extracted, two models were constructed: (1) BM-Model using radiomic features from B-mode images, and (2) CE-model using radiomic features from Sonazoid contrast-enhanced images.

#### 2.5.4. Validation of Predictive Models

The area under the receiver operating characteristic curve (AUROC) was used to evaluate each predictive model’s performance. Diagnostic metrics such as accuracy, sensitivity, and specificity were measured at the point on the ROC curve with maximum Youden’s index in the training dataset, and threshold probabilities corresponding to the point in the training dataset were used to measure the model’s diagnostic performance in the testing dataset.

#### 2.5.5. Statistical Analysis

All statistical analysis and radiomics0related model development were performed on Python (version 3.8.8) using ‘SciPy’ and ‘sklearn’ packages. Descriptive statistics are presented as the median with interquartile range or mean with standard deviation depending on the normality of statistical distribution. The normality of observations was determined using the Shapiro–Wilk test. Missing values and outliers were filled with mean or median values. Group differences were compared using Student’s *t*-test, Mann–Whitney U-test, and chi-squared test when appropriate. We calculated 95% confidence intervals (CIs) for the model’s performance metrics with the bootstrapped method. For comparison of model performance, Delong’s test was used for AUROC comparison, and the McNemar chi-squared test was used to compare accuracy, sensitivity, and specificity. A *p*-value less than 0.05 was considered statistically significant.

## 3. Results

### 3.1. Baseline Characteristics

From October 2020 to May 2021, 100 patients were prospectively enrolled in this study, including 70 patients in the training cohort and 30 patients in the validation cohort. The patients included in this study were 78.0% men (*n* = 78), with a mean age of 59 years ± 10. Baseline characteristics were not significantly different between the training and validation cohorts (Table 1). Among all 100 patients, high Ki-67 expression was pathologically diagnosed in 82 (82.0%) patients; low Ki-67 expression was pathologically diagnosed in 19 (18.0%) patients. No association between patients’ baseline characteristics and HCC histological grades was found, or with Ki-67 expression levels (Table 2).

### 3.2. Classification Performance on HCC Stage I and II versus III

Five radiomics features from BMUS and 13 from S-CEUS were finally selected using the above-mentioned feature selection method. The ROC curves of BM-Model and CE-Model in training and testing datasets are shown in Figure 2. Both models displayed good predictive value compared with random guessing (diagonal line). A higher AUROC was observed for the CE-Model on both the training and testing datasets compared with the BM-Model (0.908 vs. 0.805 in training, 0.792 vs. 0.604 in testing); the differences between AUROCs of the BM-Model and CE-Model were significant based on the results of the Delong tests (all *p* < 0.05) (Table 3).

There were significant differences in prediction accuracy (62.9% vs. 84.3% in training, 66.7% vs. 76.7% in testing) and specificity (54.5% vs. 81.8% in training, 62.5% vs. 75.0% in testing).

### 3.3. Radiomic Model Associated with Ki-67 Level

We selected 7 and 13 radiomic features for BMUS and S-CEUS images, respectively. There were no significant differences between the AUROCs of the two models in the training dataset (0.868 vs. 0.873, DeLong *p* = 0.972). However, in the testing dataset, CE-model achieved a higher and more significant AUROC of 0.768 compared to 0.712 achieved by BM-model; the difference of AUROC was significant with Delong *p* < 0.05.

There was a noticeable difference in specificity between the two models (Figure 3). CE-model achieved higher specificity of 92.3% in the training dataset and 80.0% in the testing dataset, compared to 76.9% and 60.0% by BM-model. Accuracy and sensitivity of both models displayed no significant difference (McNemar *p* = 0.12 and 0.069 in training, *p* = 0.2 and 0.834, respectively) (Table 4).

## 4. Discussion

Previous studies have demonstrated that imaging features are potentially helpful for predicting prognosis-related factors of HCC lesions, such as tumor grade, microvascular invasion, pathological subtype, and Ki-67 expression level [7,8,19]. In recent years, with the development of newer therapies for HCC, such as immune checkpoint inhibitors [21] and regorafenib [22], the radiomics approach, including texture analysis, shape, and intensity features, shows a potential correlation between medical imaging and personalized medicine [16].

Recently, Ki-67 protein expression level has been regarded as a proliferation-associated marker. The nuclear antigen was found only in proliferative cells. High Ki-67 protein expression level indicates an active status of cell proliferation. HCCs with lower Ki-67-positive tumor cell nuclei expressed high levels of VEGF, whereas those with higher Ki-67-positive tumor cell nuclei had a lower expression of VEGF [23]. Several studies have discovered the relationships between Ki-67 protein expression levels and radiomics analysis results [19,24,25]. Arterial heterogeneous hyperenhancement was found to be associated with high Ki-67 expression levels [20]. Quantitative parameters based on the signal intensity measurement of the hepatobiliary phase (HBP) of MRI preoperatively and noninvasively predicted the aggressive characteristics of HCC [17]. Some studies have used a radiomics model based on Gd-EOB-DTPA-enhanced MRI to predict Ki-67 expression in HCC, demonstrating that Ki-67 status can be predicted through radiomics analysis features. The predictive performance of the radiomics model derived from different phases of contrast-enhanced MRI was compared [20,26]. Additionally, a combined model including serum AFP level and contrast enhanced MRI scores was established and validated for predicting Ki-67 expression in HCC patients [27]. In our study, the Kupffer phase of S-CEUS radiomic features was used to predict the diagnosis of HCC pathological stages as well as Ki-67 level. The data suggested that radiomics models derived from Kupffer-phase CEUS images obtained a good result and would be feasible in clinical practice.

In this study, we analyzed the relationships between radiomics features and HCC stages, and used radiomics features to identify the more advanced stage of HCC (stage III). The developed radiomic models established a mathematical relationship between ultrasound radiomic features and pathological results. The CE-Model was constructed using quantitative radiomic features from Kupffer-phase images of Sonazoid CEUS, which display unique tissue characteristics after contrast agent uptake, to differentiate HCC at different stages. Compared with the BM-Model, which was based on regular BMUS scans, CE-Model presented a higher AUROC (0.865 vs. 0.750, *p* < 0.05) and sensitivity (88.3% vs. 66.7%, *p* < 0.05) on the differentiation of stage I and II versus stage III HCC patients in the training set, which showed the potential to assist in the diagnosis of HCC stage during ultrasound scans. A higher sensitivity was found over higher specificity because 9 out of 41 patients (22%) in the testing group were pathologically confirmed stage III HCC.

Moreover, the CE-Model demonstrated good generalizability on new tasks such as the classification of Ki-67 expression of HCC tumors (AUROC 0.768, testing dataset), while the BM-Model displayed weaker classification ability (AUROC 0.712, testing dataset). Both models displayed no significant differences in accuracy and sensitivity. Still, a higher specificity was observed for the CE-Model (92.3% and 80.0% in training and testing, respectively), which was preferable due to there being fewer patients with lower Ki-67 expression levels in the entire study group (*n* = 19, 18.8%). Interpreting the complex associations between the biological processes and radiomics features remains an enormous challenge, although it aligns with the current trend toward precise and personalized medicine [28,29].

## 5. Limitations

The main limitation of this study is that it is a single-centered, prospective study with a limited patient sample. In the future, a multicenter study with an extended population size and balanced patient distribution is necessary. The diagnostic values of the arterial, portal venous, and late phases images of CEUS also remain to be investigated.

## 6. Conclusions

The radiomic model constructed from the Kupffer phase of S-CEUS images has potential for predicting Ki-67 expression in patients with HCC. Moreover, it can provide valuable information to differentiate the histologic stages of HCC lesions.

## Figures and Tables

**Figure 1 diagnostics-12-02175-f001:**
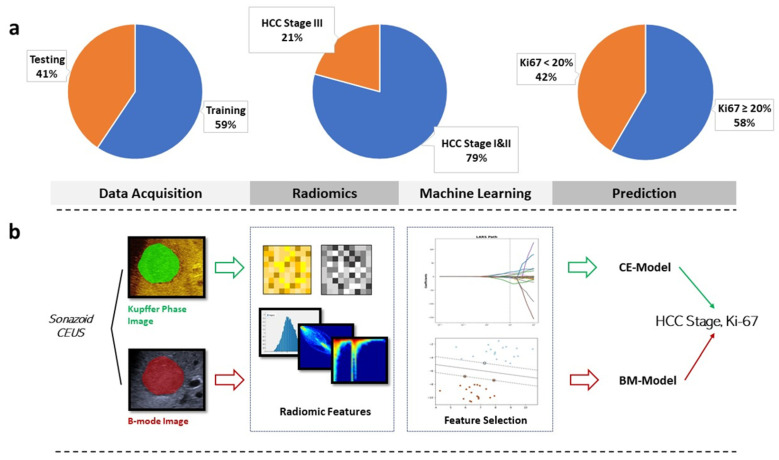
Study design. (**a**): Group size ratio of training/testing groups, different HCC stages and Ki-67 indexes. (**b**): Workflow of radiomic analysis of grayscale (B-mode) and Kupffer CEUS images.

**Figure 2 diagnostics-12-02175-f002:**
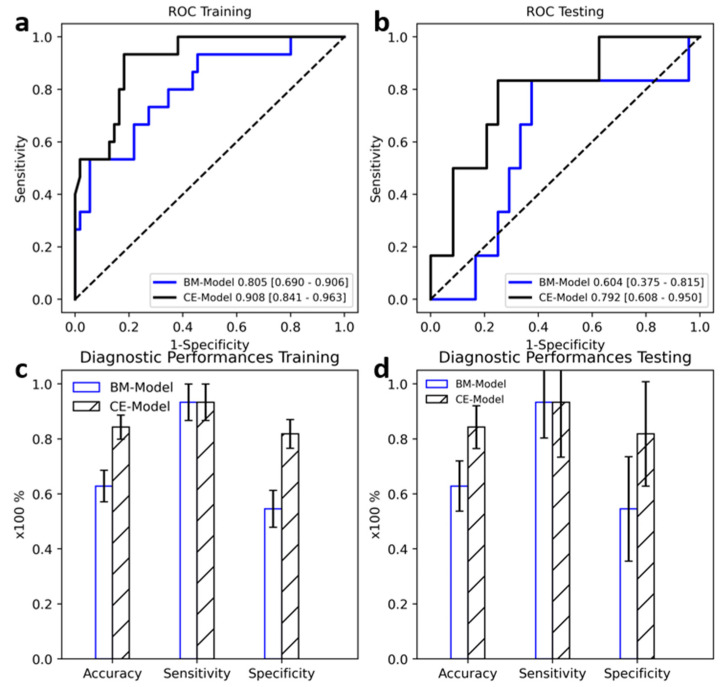
Diagnostic performance of B mode ultrasound (BM-Model) and contrast-enhanced ultrasound (CE-Model) in the training (**a**,**c**) and testing (**b**,**d**) groups. Error bar indicates 95% confidence interval.

**Figure 3 diagnostics-12-02175-f003:**
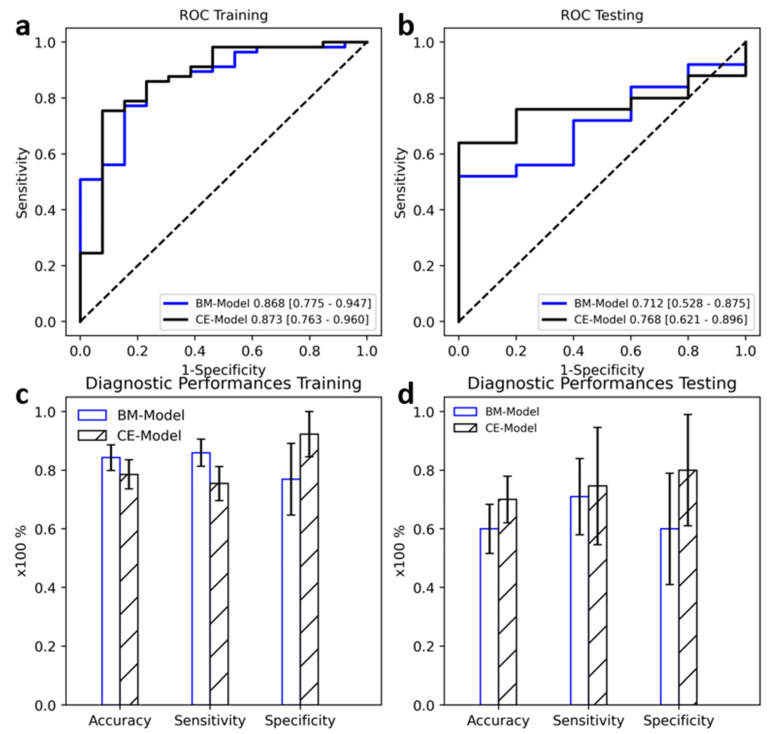
Diagnostic Performances of BM-Model and CE-Model at the prediction of Ki-67 levels. (**a**,**c**): Performances in Training Dataset. (**b**,**d**): Performances in Testing Dataset. Error bar indicates 95% confidence interval.

**Table 1 diagnostics-12-02175-t001:** Baseline characteristics of patients.

Parameter	Study Population (*n* = 101)	Training Group (*n* = 60)	Testing Group (*n* = 41)	*p*
**Sex**				0.686
male	78 (77.2)	45 (75.0)	33 (80.5)	
female	23 (22.8)	15 (25.0)	8 (19.5)	
**Age**	59.35 ± 10.07	58.63 ± 10.84	60.39 ± 8.72	0.394
**Serum Biomarkers**				
**AFP**	8.80 [2.90–46.40]	6.80 [2.68–23.95]	8.80 [3.70–94.70]	0.071
**CEA**	2.24 ± 0.94	2.26 ± 0.96	2.20 ± 0.91	0.742
**CA19-9**	12.20 [7.60–14.90]	10.90 [7.60–13.85]	12.20 [7.90–15.60]	0.233
**AFP-L3**	5.30 [0.50–8.75]	4.80 [0.50–6.65]	5.30 [0.50–13.20]	0.134
**PIVKA**	93.00 [29.00–131.00]	93.00 [28.50–132.75]	39.00 [29.00–123.00]	0.272
**miRNA**				
positive	76 (75.2)			
negative	25 (24.8)			
**HCC Stage**				0.010
I and II	80 (79.2)	48 (80.0)	32 (78.0)	
III	21 (20.8)	12 (20.0)	9 (22.0)	
**Ki-67**				0.051
<20%	42 (41.6)	23 (38.3)	19 (46.3)	
≥20%	59 (58.4)	37 (61.7)	22 (53.7)	

Descriptive statistics are presented as *n* (%) for categorical variables, mean ± standard deviation for normally distributed variables, and median [interquartile range] for nonuniformly distributed variables. CEA, carcinoembryonic antigen; CA19-9, carbohydrate antigen 19-9; AFP-L3, alpha-fetoprotein isoform L3; PIVKA-II, protein induced by vitamin-K deficiency II; HCC, hepatocellular carcinoma.

**Table 2 diagnostics-12-02175-t002:** Univariate logistic regression revealed no clinical data were significantly associated with HCC staging or Ki-67 level.

	HCC Staging				Ki-67 Level			
Variable	OR	[0.025	0.975]	*p*	β	[0.025	0.975]	*p*
**Sex**	0.328	0.086	1.268	0.106	0.625	0.183	2.125	0.452
**Age**	0.923	0.870	0.980	0.108	0.954	0.904	1.007	0.090
**AFP**	1.000	1.000	1.000	1.000	1.002	0.998	1.005	0.297
**CEA**	0.782	0.452	1.355	0.382	0.944	0.582	1.532	0.817
**CA19-9**	1.007	0.970	1.044	0.700	0.965	0.932	1.000	0.051
**AFP-L3**	1.023	1.005	1.041	0.240	1.010	0.991	1.029	0.281
**PIVKA**	1.000	1.000	1.000	1.000	1.000	1.000	1.000	1.000
**miRNA**	2.333	0.456	11.917	0.309	1.555	0.467	5.181	0.472

CEA, carcinoembryonic antigen; CA19-9, carbohydrate antigen 19-9; AFP-L3, alpha-fetoprotein isoform L3; PIVKA-II, protein induced by vitamin-K deficiency II.

**Table 3 diagnostics-12-02175-t003:** Diagnostic performance of BM-Model and CE-Model.

	Data Group	AUROC	Accuracy	Sensitivity	Specificity
**Grayscale**	Training	0.805 [0.690–0.906]	0.629 [0.529–0.729]	0.933 [0.812–1.000]	0.545 [0.434–0.655]
	Testing	0.604 [0.375–0.815]	0.667 [0.367–0.667]	0.833 [0.500–1.000]	0.625 [0.250–0.583]
**Kupffer CEUS**	Training	0.908 [0.841–0.963]	0.843 [0.771–0.914]	0.933 [0.812–1.000]	0.818 [0.727–0.900]
	Testing	0.792 [0.608–0.950]	0.767 [0.633–0.900]	0.833 [0.333–1.000]	0.750 [0.640–0.920]
**Difference *p***	Training	0.015	0.020	1.000	<0.001
	Testing	<0.001	0.042	0.969	0.021

96% confidence intervals are included in parenthesis. *p*-value was calculated with the DeLong test for AUROC comparison and a two-sided permutation test with 10,000 random resamples.

**Table 4 diagnostics-12-02175-t004:** Diagnostic Performance of Radiomic Models on Ki-67 level Prediction.

	Data Group	AUROC	Accuracy	Sensitivity	Specificity
**Grayscale**	Training	0.868 [0.775–0.947]	0.843 [0.771–0.914]	0.860 [0.778–0.931]	0.769 [0.556–0.944]
	Testing	0.712 [0.528–0.875]	0.600 [0.567–0.833]	0.710 [0.615–0.893]	0.600 [0.000–0.800]
**Kupffer CEUS**	Training	0.873 [0.763–0.960]	0.786 [0.700–0.871]	0.754 [0.655–0.847]	0.923 [0.778–1.000]
	Testing	0.768 [0.621–0.896]	0.700 [0.600–0.867]	0.746 [0.667–0.920]	0.800 [0.500–0.833]
**Difference *p***	Training	0.873	0.12	0.069	<0.001
	Testing	0.048	0.2	0.834	<0.001

96% Confidence Intervals were included in parenthesis. *p*-value was calculated with the DeLong test for AUROC comparison and a two-sided permutation test with 10,000 random resamples.

## Data Availability

Data available on request due to restrictions ethical. The data presented in this study are available on request from the corresponding author. The data are not publicly available due to restrictions ethical.

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
