# Peer review of "Prediction of Histological Grades and Ki-67 Expression of Hepatocellular Carcinoma Based on Sonazoid Contrast Enhanced Ultrasound Radiomics Signatures"

_diagnostics, 2022, doi:10.3390/diagnostics12092175_

Round 1

Reviewer 1 Report

Interesting paper. My main concern is on the real utility of Ki-67 assessment in HCC patients, being this marker important specifically in patients with NETs. The authors should clarify this important point.

I would suggest to quickly comment in the discussion the impact of these findings in light of the newer therapies for HCC, such as immune checkpoint inhibitors (cite the paper PMID: 33086471) and regorafenib (cite PMID: 31877664)

Author Response

Dear Reviewer,

Thank you for the comments concerning our manuscript entitled “Prediction histological grades and Ki-67 expression of hepatocellular carcinoma based on Sonazoid contrast enhanced ultrasound radiomics signatures” (Manuscript ID diagnostics-1891358). The comments have been important and helpful to improve our paper. All changes were highlighted by tracking mode in our revised manuscript. Please find our point-to-point reply below.

Thank you with Best regards!

Prof. Dr. med. Wen-Ping Wang

Point 1: My main concern is on the real utility of Ki-67 assessment in HCC patients, being this marker important specifically in patients with NETs. The authors should clarify this important point.

Response 1: Previous studies showed that Ki-67 protein expression is an independent indicator of tumor growth rate and poor prognosis in HCC. Also, the Ki-67 expression reflects the degree of tumor proliferation activity and is significantly correlated with tumor grades. We discussed this in the introduction part with citation of literatures.

Point 2: I would suggest to quickly comment in the discussion the impact of these findings in light of the newer therapies for HCC, such as immune checkpoint inhibitors (cite the paper PMID: 33086471) and regorafenib (cite PMID: 31877664).

Response 2: The comment were added to the discussion part with citation of these important papers.

Reviewer 2 Report

In this study, the authors proposed a prediction model for tumor histopathological grades and Ki-67 expression in patients with hepatocellular carcinoma. The idea is to integrate radiomics features and machine learning to create a final prediction model. Although the results look promising, some major points should be addressed as follows:

1. It is a kind of prospective study, thus I suggest the authors split the data into training and validation using some strategies (not random). For example, using the retrieved date as criteria to split.

2. Order of feature selection techniques is a concern also. Why did the authors use mRMR before applying LASSO? If LASSO is applied before mRMR, will the results be different?

3. Which statistical tests were applied to get the error bars (in some figures i.e., Fig. 2b)?

4. The authors should compare their predictive performance to previously published works on the same problem/data.

5. Sometimes their model reached a final AUC at 0.661 (Ki-61 level prediction). It is not a quite good performance, so how to know that the authors have created a good model in this case?

6. It is suggested to perform cross-validation in the training process.

7. In the "Methodology" section, the authors should add more descriptions related to their machine learning model implementation.

8. Measurement metrics (i.e., sensitivity, accuracy, AUC, etc.) are well-known and have been used in previous biomedical studies such as PMID: 34502160, PMID: 34915158. Thus the authors are suggested to refer to more works in this description to attract a broader readership.

9. Quality of figures should be improved.

10. English writing and presentation style should be improved as well.

Author Response

Dear Reviewer,

Thank you for the comments concerning our manuscript entitled “Prediction histological grades and Ki-67 expression of hepatocellular carcinoma based on Sonazoid contrast enhanced ultrasound radiomics signatures” (Manuscript ID diagnostics-1891358). The comments have been important and helpful to improve our paper. All changes were highlighted by tracking mode in our revised manuscript. Please find our point-to-point reply below.

Thank you with Best regards!

Prof. Dr. Wen-Ping Wang

Point 1: It is a kind of prospective study, thus I suggest the authors split the data into training and validation using some strategies (not random). For example, using the retrieved date as criteria to split.

Response 1: It is indeed very reasonable to split the data into training and testing groups using data retrieval data and criteria. We have changed the splitting criteria to the patient admission date and updated the result. In terms of model performance on newer datasets, there was no significant deviation from the previous result.

Point 2: The order of feature selection techniques is a concern also. Why did the authors use mRMR before applying LASSO? If LASSO is applied before mRMR, will the results be different?

Response 2: The reason behind applying mRMR before LASSO is mRMR ranks features in terms of their relevance with the prediction target (F-statistics) and intra-feature correlation (Pearson’s Correlation Coefficient) to reduce the original feature set to a smaller set of features, while LASSO uses penalization parameter in a logistic regression which is like the final form of our model. This order of machine learning feature selection applications was also seen in other works such as (PMID: 32010566) and (PMID 32820210). We applied LASSO prior to mRMR and used stepwise regression to formulate an alternate model, the resulting final features were not the same and the model performance was not as excellent.

Point 3: Which statistical tests were applied to get the error bars (in some figures i.e., Fig. 2b)?

Response 3: The error bars in the figures indicate 95% confidence intervals, calculated using a bootstrapped method. We have not made it clear in the paper, it will be  now noted in the figure description section.

Point 4: The authors should compare their predictive performance to previously published works on the same problem/data.

Response 4: Previously, there is no similar paper published on radiomic model constructed from Kupffer phase of S-CEUS images in predicting Ki-67 expression in patients with HCC. Previously, some studies used a radiomics model based on contrast-enhanced MRI to predict Ki-67 expression in HCC demonstrate that Ki-67 status. We cited and discussed them in our discussion part.

Point 5: Sometimes their model reached a final AUC at 0.661 (Ki-61 level prediction). It is not a quite good performance, so how to know that the authors have created a good model in this case?

Response 5: The updated result using admission date as splitting criteria showed improved performance on Ki-67 level prediction.

Point 6: It is suggested to perform cross-validation in the training process.

Response 6: In the updated result, feature selection and model tuning were cross-validated (10-folds). LASSO penalty factor were tuned using 5-fold cross-validation.

Point 7: In the "Methodology" section, the authors should add more descriptions related to their machine learning model implementation.

Response 7: We have added additional descriptions regarding feature selection and model constructions.

Point 8: Measurement metrics (i.e., sensitivity, accuracy, AUC, etc.) are well-known and have been used in previous biomedical studies such as PMID: 34502160, PMID: 34915158. Thus the authors are suggested to refer to more works in this description to attract a broader readership.

Response 8: Thank you for your suggestion. These two papers were cited in our revised manuscript accordingly.  

Point 9: Quality of figures should be improved.

Response 9: Figures were reuploaded for our revision. Hopefully they are better improved this time.  

Point 10: English writing and presentation style should be improved as well.

Response 10: A native speaker (Prof. Dietrich) helped us to improve the English writing and presentation. Hopefully they are better improved this time.

Round 2

Reviewer 2 Report

My previous comments have been addressed.